# Molecular Diagnosis of COVID-19 Sudden and Unexplained Deaths: The Insidious Face of the Pandemic

**DOI:** 10.3390/diagnostics13182980

**Published:** 2023-09-18

**Authors:** Dagmara Lisman, Grażyna Zielińska, Joanna Drath, Aleksandra Łaszczewska, Ilona Savochka, Mirosław Parafiniuk, Andrzej Ossowski

**Affiliations:** 1Forensic Genetic Department, Pomeranian Medical University, 70-204 Szczecin, Poland; grazyna.zielinska@pum.edu.pl (G.Z.); joanna.drath@pum.edu.pl (J.D.); aleksandra4027@gmail.com (A.Ł.); andrzej.ossowski@pum.edu.pl (A.O.); 2Forensic Medicine Department, Pomeranian Medical University, 70-204 Szczecin, Poland; ilona.savochka@pum.edu.pl (I.S.); miroslaw.parafiniuk@pum.edu.pl (M.P.)

**Keywords:** 2019 novel coronavirus disease, COVID-19, RNA stability, RT-PCR, SARS-CoV-2, coronavirus, coronavirus disease, infectivity, postmortem, respiratory infections, viruses, deaths sudden

## Abstract

The COVID-19 epidemic has led to a significant increase in the number of deaths. This has resulted in forensic autopsies focusing on additional diagnostic possibilities. The following article is a summary of 23 autopsies of sudden and unexplained deaths. Particularly noteworthy are the described cases of children whose deaths were originally classified as SIDS (sudden infant death syndrome). All tests were performed at the Department of Forensic Medicine and Forensic Genetics, Pomeranian Medical University in Szczecin. Autopsy analyses were extended to include diagnostics of the SARS-CoV-2 virus using molecular methods and a detailed histopathological analysis of lung tissue. The material for molecular tests consisted of a nasopharyngeal swab taken postmortem and a lung tissue homogenate. In both cases, the RT-PCR method with CT cut-off point analysis was used for diagnosis. In all analyzed cases, the lungs showed massive congestion and increased fragility and cohesion. The tested material showed the presence of the SARS-CoV-2 virus, which indicated various stages of infection. It was observed that the higher the virus expression in the lungs, the lower or undetectable it was in the nasopharyngeal swab. This may explain false negative results during life in swabs. An interesting finding is that child deaths classified as SIDS also showed the presence of the virus. This may constitute a new direction of research.

## 1. Introduction

Most infections with Coronaviridae viruses are mild, but they can be responsible for respiratory diseases in the form of severe respiratory syndromes with high mortality. The first cases of complicated pneumonia were reported in December 2019 in China (Wuhan, Hubei), now known as COVID-19 disease caused by the SARS-CoV-2 virus. This virus has spread around the world relatively quickly [1]. In January 2020, the WHO (World Health Organization) declared a Public Health Emergency of International Concern (PHEIC), followed by a pandemic in March 2020 [2].

So far, several postmortem examinations conducted during the pandemic (from the beginning of 2019) have been presented [3,4,5,6,7,8,9,10,11,12,13,14,15,16,17,18,19,20,21]. These analyses still provide little information about the pathomechanism of the COVID-19 disease and the mechanism of death itself, and most importantly, they do not accurately describe the causes of death among children. Initially, COVID-19 was described as an entity attacking only the respiratory system. Only the autopsy, combined with clinical data, showed that it was a multi-organ dysfunction [15,22].

Autopsy studies should pay special attention to the condition of the internal organs and the patient’s clinic. In the case of forensic autopsies, pathologists do not have the clinical data of the deceased. Without disclosing the involvement of people who can cause death, they thoroughly analyze the internal organs. The research conducted by our team contributes to the explanation of the factors leading to death. Forensic dissections are definitely different from typical clinical dissections. The analyzed court deaths are usually sudden, and often the circumstances are unclear.

Histological examinations of individual organs show structural changes that could lead to death [23]. In the case of autopsies performed at the Department of Forensic Medicine, their nature is slightly different. An autopsy is carried out at the prosecutor’s request. The medical examiner must preliminarily exclude or confirm the involvement of third parties. Autopsies analyzed in this study were performed at the Department of Forensic Medicine in Szczecin. The corpses of the persons subjected to autopsies were handed over to the ZMS Prosecutor’s Office at the request of the Prosecutor’s Office. Deaths occurred suddenly, and bodies were revealed in apartments or other circumstances.

There are works that describe the presence of the SARS-CoV-2 virus in people infected for more than five months [24], but they do not provide an answer regarding the exact time for which the virus is able to survive in the human body. The mere fact of detecting the presence of the genome of this virus does not provide an answer as to whether it is alive and whether it is an etiological agent. These may be non-infectious short fragments of its genome. The biggest problem is to determine its survival time in the corpse and the source of infection for the pathologist. Researchers have studied several viruses so far. Studies on the analysis of influenza RNA genomes provide information that they can remain detectable in frozen bodies for decades [25]. The SARS-CoV-2 virus, like the influenza virus, is also a single-stranded RNA virus, having an envelope in common with most viruses. The factors that can affect the viability of the virus are temperature and humidity. The Ebola virus, which is also an RNA virus, remains easily detected (as alive) in test samples for more than seven days after death, and its genome about 10 weeks after death. These studies were conducted in West Africa on macaques infected with the Ebola virus [26]. Postmortem analyses were also conducted on the survival time of SARS-CoV-2. All organs (heart, lungs, kidneys) secured during the autopsy were analyzed for the presence of this virus. Diagnostics were made by RT-PCR. The presence of the N and RdRp genes in the lungs was revealed, with a CT of 31 for both genes; in the heart, for the N gene, the CT was 36; genes were not detected in both kidneys. It is difficult to estimate how long the SARS-CoV-2 virus genomes remain active after death, but it is known that the viral genome persists in tissues for over 30 days [24,25]. It is difficult to estimate the time of the SARS-CoV-2 virus genome, which is active after death, but it is known that the viral genome takes more than 30 days to emerge [26,27]. This suggests the hypothesis that its transmission from the deceased is possible for some time after death, despite the observance of all safety procedures applied during autopsy [26]. The risk of infection transmission is much higher in the case of positive RNA strand polarity due to the specific structure of the virus genome, which determines its infectivity. One such virus is SARS-CoV-2. Its positive RNA strand polarity means that it can be directly mRNA, and the very isolation of such a genome determines its infectivity [27]. This allows us to consider the legitimacy of testing with a molecular test (RT-PCR) of each body delivered to the Department of Forensic Medicine, especially during a pandemic.

Such postmortem virological tests will allow us to observe the migration of the SARS-CoV-2 virus in the body. Consideration should also be given to the risk of cadaver transmission, whether for the autopsy staff or for the bereaved family. Another aspect of our research is a completely different view of sudden deaths of children, previously described as SIDS (Sudden Infant Death Syndrome). The cause of SIDS is still not fully understood. There is talk of its connection with a certain immaturity of the child’s respiratory system. Or maybe this failure is superimposed by viral infections such as COVID-19?

## 2. Material and Methods

In the period from 2020 to May 2022, 23 autopsies of people aged 2 months to 84 years (9 women and 14 men) were performed at the Department of Forensic Medicine in Szczecin. In addition to the classic forensic autopsy, these people underwent diagnostics for the presence of genetic material of the SARS-CoV-2 virus RNA. Diagnostics were performed by RT-PCR (Polish reverse transcription polymerase chain reaction). During the dissection, standards and protocols for the prevention of infectious diseases were used. Histopathological analyses were also performed. COVID-19 postmortem diagnosis was based on the results of molecular tests obtained by RT-PCR with the analysis of CT cut-off points (cycle threshold value) and the SARS-CoV-2 virus sequence in postmortem nasopharyngeal swabs and lung tissue homogenates. An internal control ensured that the samples were successfully isolated and amplified.

Samples with a detected result for all three or two genes were interpreted as SARS-CoV-2 PCR positive, according to protocol. Clinical data of the deceased were not available. Samples showing amplification of the internal control but no detection of the target genes were classified as negative. Samples not showing amplification of the internal control were considered invalid.

Demographic data (age, sex) are presented in Table 1. Presents the results of the RT-PCR molecular test, taking into account the CT cut-off points. All autopsy authorization documents were delivered to the ZMS prior to the autopsy. Postmortem interval (PMI) to autopsy was 5 to 30 days.

The material for genetic testing was a swab from the nasopharynx taken postmortem and a fragment of the pathologically changed lung tissue. Nasopharyngeal material was collected on sterile nylon swabs with flexible plastic shafts. After collection, the swab was placed in 500 µL of the transport medium (R9 A&A), which has the ability to lyse and inactivate the virus until its isolation. This prevents possible contamination of personnel during sample transport. The material collected in this way was transported as soon as possible to the Department of Orchard Genetics, where molecular diagnostics for the presence of SARS-CoV-2 were performed.

A piece of lung tissue (approximately 1 cm × 1 cm) was also placed in 500 µL of the transport medium (R9 A&A). The tissue was homogenized in a manual homogenizer using inert glass beads immediately prior to isolation.

The sample was digested for 10 min. Prior to direct isolation, each sample was vortexed for 5 s. Automatic isolation on the King Fisher analyzer (Applied Biosystems, Waltham, MA, USA) was performed using the MagMax Viral Pathogen II (MVP II) Nucleic Acid Isolation Kit (Applied Biosystems) [28], according to the manufacturer’s protocol. To verify the correctness of the reaction, the TaqPath COVID-19 CE-IVD RT-PCR Kit (Applied Biosystems) [29] was used to perform the RT-PCR. This kit is dedicated to nasopharyngeal swabs and tissues and contains reagents and controls for the reaction RT-PCR Designed for the qualitative detection of SARS-CoV-2 virus nucleic acids. The kit contains three sets of primers and probes for identifying specific genomic regions of the SARS-CoV-2 virus (ORF, N, S) and a set of primers and probes specific for the SARS-CoV-2 virus human RNA-se P gene (RPPH1) as an internal control Probes bind to three viral target sequences: ORF 1ab gene, N protein, and S protein.

In the final stage, the obtained results were analyzed using the QuantStudio5 Analysis software v1.2 (Applied Biosystems).

## 3. Results

### 3.1. Pathology Findings

The lungs were macroscopically changed and showed increased cohesiveness. In practically all cases, the lung parenchyma was massively hyperaemic, showing increased fragility on palpation. On the surface of the lungs, there was a slight localized dark-colored edema (dotted pattern) characteristically intersecting the surfaces (Figure 1), with small hemorrhages in some places (Figure 2).

The histological picture of the lung tissue was dominated by massive hyperemia and edematous changes in the alveoli and dilation of the capillary vessels. Diffuse Alveolar Damage (DAD) was generally observed. Fibrin deposits and clusters of macrophages were also visualized (Figure 2) as extravasation of blood from the airspace of the lungs. The alveoli showed signs of fibrosis in some cases (Figure 3).

Such changes were observed in the acute phase of infection. Most likely, it was the cause of respiratory failure (24 h after the infection was detected).

These changes may also indicate a chronic inflammatory process. These processes result in the formation of a characteristic lung tissue that is compared to a honeycomb. Molecular tests confirmed infection with the SARS-CoV-2 virus. Therefore, we can say that chronic bilateral pneumonia was detected in all the deceased. These are cases in which productive changes dominated and occurred in patients a few weeks after infection, leading to respiratory failure. The mechanism of death was different than in the acute phase of COVID-19.

Changes were revealed during histopathological examinations. Samples were taken only from macroscopic solutions. The collected lungs were fixed in 10% buffered formalin and then embedded in paraffin (FFPE). The blocks were cut on a microtome, stained (H-E, hematoxylin-eosin), and then the preparation was viewed under magnification. Various images were subjected by appropriate pathologists to Zeiss, which allows for further subtle structure changes (Filter set 25 Zeiss parameters, Excitation TBP 400 + 495 + 570, Emission TBP 460 + 530 + 625).

### 3.2. Genetic Results

The results of genetic tests by RT-PCR are presented in Table 1.

The tested samples from lung tissue homogenates and postmortem nasopharyngeal swabs showed the presence of SARS-CoV-2 virus RNA. The CT cycle threshold refers to the number of cycles needed to amplify the viral RNA to reach a detectable level. A high CT value may not accurately identify a patient’s infectious potential [29,30,31]. We used the CT value as an indicator of a positive result, which may suggest the spread of the virus in the body. The average time from death to autopsy was about 14 days. The samples were found to be positive for the presence of SARS-CoV-2 virus RNA genetic material in all analyzed cases, despite the fact that the bodies were kept in a cold store at 4 °C until the autopsy was performed. Such a temperature for storing a corpse ensures that the progress of decomposition of the body and the progressing putrefaction processes are inhibited.

Four of the analyzed cases were deaths of children aged 2 months to 7 years. So far, it has been said that children are susceptible to SARS-CoV-2 infection, but the course of the disease is mild [32]. This approach meant that children under the age of 12 were practically not tested as often as adults. The case of a 7-year-old girl we analyzed shows that this approach is not correct. The child was in quarantine with his mother, who had a positive RT-PCR test for SARS-CoV-2. During the approximately 14-day quarantine period, the child tested negative in the RT-PCR test. About 30 days after the confirmation of the mother’s infection, the girl, who did not show any clinical symptoms, died in the morning. Characteristic lung changes and molecular testing showed the presence of genetic material of the SARS-CoV-2 virus both in the lungs and in the nasopharynx, with the absence of the S gene in the nasopharyngeal swab.

Analyzing the case of a 2-month-old child, the results of nasopharyngeal swabs confirmed the presence of the virus for all tested genes (N, S, and ORF). Lack of expression in the lungs may indicate the initial stage of SARS-CoV-2 infection. A similar relationship occurs in the case of a 14-month-old child. The effectiveness of RT-PCR tests is closely related to the material analyzed and the time that has elapsed since the first symptoms appeared. After 10 days from the onset of symptoms, nasopharyngeal swab results may be false negative, and the chances of a positive result decrease [33]. Expression in both the lungs and the nasopharynx is present in the material collected from a 7-year-old and a 3-month-old child. In both of these cases, the ORF gene was not detected in the nasopharynx. The presence of the SARS-CoV-2 virus in the lungs may indicate the severity of COVID-19.

In eight cases, one of the genes was “dropped out”, including the ORF gene in seven cases and the S gene in one case. This is due to the mutation of the SARS-CoV-2 virus. In the case of new mutations, they have a 70% greater infection capacity. This is because the human body begins to produce antibodies after entering the virus. Therefore, the virus begins to change the composition of its envelope to avoid recognition by host cells. This may be related to the latent course of the disease, which in turn leads to death.

In nine analyzed cases (samples No. 6, 7, 9, 10, 16, 17, 20, 21, and 23), the genetic material of the SARS-CoV-2 virus was not detected in the nasopharynx, but it was detected in the lung tissue homogenate. The CT averaged 25 to 37. This may indicate an ongoing chronic COVID-19 infection. In such cases, the patient does not have any characteristic symptoms of the disease, or they are few (runny nose, slight cold). Chronic infection, as in the analyzed cases, leads to death. Such a picture of the disease may result in the patient obtaining a false negative RT-PCR test result for the SARS-CoV-2 virus while there is still an active infection in the lung tissue, which can be fatal. In the case of sample No. 12, the genetic material was detected only in the nasopharyngeal swab, and it was only the ORF and N gene. This may indicate an active infection. In the remaining cases, genetic material was detected in both the nasopharynx and the lung tissue. Other researchers have reached similar conclusions [34,35]. Their research confirms our analysis that the SARS-CoV-2 virus is detected postmortem.

## 4. Discussion

Our observations allow us to conclude that in all cases, the SARS-CoV-2 virus was the cause of death. The obtained molecular test results and histopathological analyses indicated COVID-19 disease. In the case of infants, there is a suspicion that a chronic viral infection could overlap with SIDS, which is confirmed by the characteristic lung picture in the described cases of infants. Autopsies of elderly people also showed a picture of a chronic inflammatory process and even fibrosis of the lung tissue. There are studies on the impact of the SARS-CoV-2 virus on the heart muscle [36].

The pandemic has significantly overburdened the global health service and the medico-forensic community [37,38]. In references [30,31] This contributed to the fact that the Italian Ministry of Health issued guidelines on the waiver of autopsies of persons suspected of COVID-19 [39,40]. Autopsy studies are extremely important in the cognitive process of pathophysiology. They also proved valuable in understanding the mechanism of death as a result of COVID-19 [41]. They allowed to prevent and control the epidemic [26,33]. They also contributed to understanding the processes of clotting disorders observed in COVID-19 [42].

In references [24,25,27,29,30,37,38,43] Previous autopsy studies of people who died from COVID-19 showed the presence of the virus’s genetic material in the corpses lasting up to several days [44]. Studies similar to ours showed that the virus persisted for up to 35 h in nasopharyngeal swabs secured by postmortem [19]. Our study confirms these observations.

ICMR (Indian Council of Medical Research) has defined a basic CT range to facilitate the diagnosis of SARS-CoV-2 infection. For values greater than 35 CT, the result is considered negative. A value between 25–35 CT indicates moderate viral load and transmissibility; values below 25 CT suggest high viral load and a strong predisposition to transmission [45]. The standard for COVID-19 diagnosis is the RT-PCR test, most often performed with a nasopharyngeal swab. However, false negative results may occur. In our study, analyzing the case of a 67-year-old man, the swab from the nasopharynx gave a negative result, while the swab from the lungs of the same person was positive. This may indicate that the absence of the SARS-CoV-2 virus on the mucous membranes of the nose and throat does not mean that there is no infection. Therefore, the patient may receive a false negative result during his lifetime. The concentration of viral RNA in the upper respiratory tract peaks shortly before or just after the onset of symptoms and also fluctuates throughout the day. In the later stages, the infection decreases exponentially [37]. Within the first few days after the onset of infection, viral multiplication in the upper respiratory tract reaches its highest values. Within the larynx, trachea, and bronchi, independent replication also occurs. SARS-CoV-2 is detectable from a few days to a few weeks after the first symptoms appear [38]. A 2020 study showed lower sensitivity for nasal (63%) and throat (32%) swabs than for bronchoalveolar lavage (93%) and sputum (72%) [37,45]. Repeatedly negative throat/nose swab tests are not a diagnostic certainty in the case of high clinical suspicion of SARS-CoV-2 infection. There is a correlation between a positive RT-PCR test using bronchoalveolar lavage (73%) and CT scans characterized as images typical of COVID-19. This raises doubts about the effectiveness of using nasopharyngeal swabs as a standard for RT-PCR diagnostics. However, the sensitivity of upper and lower respiratory tract tests is dependent on the time since infection. In the early stages of the disease, swabs from the upper respiratory tract are more sensitive; in the more advanced stage, the lower respiratory tract is characterized by this feature [38]. The results recorded in the COVID-19 advanced lung swab, in the later signal course, no positive signal for the nasopharynx is correct. The use of the bronchoalveolar procedure as a diagnostic standard is problematic due to the procedure of the invasive procedure [30].

In eight of the twenty-three cases analyzed, the phenomenon of gene “dropping out” was noted. Coronaviruses have a wide spectrum of genetic plasticity. Genetic mutations affect virulence and susceptibility to transmission. SARS-CoV-2 belongs to RNA viruses that are characterized by a high predisposition to mutation. It is estimated that more than 56,000 mutations, including deletions and insertions, have arisen since the emergence of SARS-CoV-2. The most common mutations concern the ORF1ab gene—71.3%, the S (spike) gene—12.8%, and the N (nucleocapsid) gene—4.2% (34). E, RdRp, N, and ORF1ab genes are targets for commercial and laboratory test primers and probes [31].

In the presented study, different expression of the virus in nasopharyngeal swab and lung tissue is observed. In some patients, it is present in both cases, and in others, it is found in either a nasopharyngeal swab or lung tissue. Detection of the virus in the lung epithelial cells as well as in the respiratory tract is possible during the acute phase of lung damage in patients with respiratory failure. A similar relationship does not occur with pneumonia in the phase characterized by inflammation of the tissue filling the alveoli and bronchioles [29]. We observed such a picture of the lungs in our study. It was especially pronounced in children. Pediatric Ct scores range from 9 to 31 Ct. Children are exposed to a high probability of SARS-CoV-2 infection and a course characterized by less intense symptoms than adults. There was a suspicion that low levels of the SARS-CoV-2 virus in children contribute to false-negative RT-PCR results, and undiagnosed pediatric patients are a reservoir of the germ [32].

A study from 2021 [29] investigated the effectiveness of RT-PCR diagnostics in children with low titers of the SARS-CoV-2 virus. The material was taken from the nasopharynx of children and adults with a median of 6 days from the onset of symptoms. RT-PCR results for children were negative, but serological tests performed after six weeks indicated SARS-CoV-2 infection [29]. However, there is insufficient evidence to support the relationship between age and viral load. CT values for children are comparable to those for adults [44]. An early 2020 study [32] also confirms the similar nature of the results in adults and children and also points to the recording of much higher amounts of virus genetic material in children under 5 years of age [32]. The concentration of viral RNA in the upper respiratory tract peaks shortly before or just after the onset of symptoms, and also fluctuates throughout the day. In the later stages, the infection decreases exponentially [37]. Sputum or bronchoalveolar lavage samples from living individuals may have increased diagnostic efficacy. Within the first few days after the onset of infection, viral multiplication in the upper respiratory tract reaches its highest values. Within the larynx, trachea, and bronchi, independent replication also occurs. SARS-CoV-2 is detectable from a few days to a few weeks from the moment the first symptoms appear [46]. A 2020 study showed lower sensitivity for nasal (63%) and throat (32%) swabs than bronchoalveolar lavage (93%) and sputum (72%) [37]. This may be the reason for obtaining false positive results in patients diagnosed with symptoms of the disease. This raises doubts about the effectiveness of using nasopharyngeal swabs as a standard for RT-PCR diagnostics. However, the sensitivity of upper and lower respiratory tract tests is dependent on the time since infection. In the early stages of the disease, swabs from the upper respiratory tract are more sensitive; in the more advanced stage, the lower respiratory tract is characterized by this feature [38] Positive results recorded in a swab from the nasopharynx indicate the severity of the COVID-19 disease; in the later stages of infection, a negative result for the nasopharynx is normal. However, the use of bronchoalveolar fluid as a diagnostic standard is problematic due to the need to perform an invasive procedure requiring the presence of qualified personnel [30].

Bearing in mind the above data and their potential importance based on the analysis of performed autopsies, we claim that postmortem molecular diagnostics is of great importance, if only because we still detect the genetic material of the virus in the body, which can be potentially infectious. The aim of our research was to assess the usefulness of molecular tests for SARS-CoV-2 postmortem. The test confirmed the presence of genetic material of the postmortem virus, which could pose a potential risk of infection. Different CT values were observed for the nasopharyngeal swab and lung tissue. It is also worth emphasizing that the nature of these deaths was not clear. At the time of autopsy, we did not have any clinical data that would indicate the presence of COVID-19 disease in the deceased. Only the inflammatory picture of the lung tissue, repeated in all analyzed cases, prompted us to perform molecular diagnostics. This shows how little we still know about this virus. How insidious the course of infection can be. How much the condition of a patient who may not even know about the infection can worsen.

In times of ongoing risk, strict adherence to COVID-19 safety protocols and standard procedures is essential [47]. The results of our study show that it is worth conducting postmortem molecular diagnostics towards COVID-19 because it will allow us to look more broadly at the cause of death as well as the further spread of the virus. Limiting the sharing of infected bodies with family and expediting burial will greatly minimize this risk. Comparing the results of molecular tests with the macroscopic and histopathological images of tissues seems to be necessary and should continue to be the subject of research. The presence of this virus in a corpse stored in a cold store for a relatively long time after death proves the high resistance and virulence of the SARS-CoV-2 virus. However, we would like to point out that the obtained positive results of the molecular test are not indications for routine postmortem diagnostic procedures and do not always define the main cause of death. Therefore, they should not be the only source of analyses but should only supplement histological and toxicological tests, for example, for the purpose of providing opinions in criminal cases.

COVID-19 is an insidious disease entity, as a result of which the clinical picture of the patient changes in a short time. The observed massive hyperemia of the lung tissue as a result of chronic inflammation causes a significant expansion of the capillaries, reduction of their peripheral tone, slowing down of blood flow, and consequently stopping gas exchange and death. We suggest that molecular diagnostics of the SARS-CoV-2 virus postmortem should complement the entire autopsy analysis. The deaths of children provide information that the COVID-19 disease in children has a completely different course than initially assumed. The study led to yet another reflection. A similar picture is also observed in cases of SIDS in infants. Verification of this hypothesis requires further extensive research in a wider scope.

## 5. Conclusions

The study observed variations in virus expression in nasopharyngeal swabs and lung tissue. Some patients had the virus in both, while others had it in only one of them, especially during the acute phase of lung damage. This phenomenon was particularly prominent in children, who have a high likelihood of infection but often exhibit milder symptoms. The low levels of the virus in children could lead to false-negative test results, potentially making them a source of undiagnosed infections.

## Figures and Tables

**Figure 1 diagnostics-13-02980-f001:**
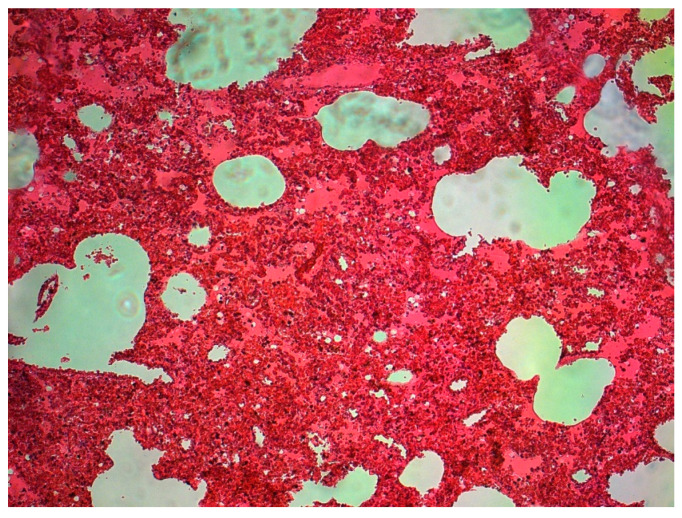
Massive congestion of the lung tissue, dilated capillaries—a child aged 7 (source: Prof. Mirosław Parafiniuk).

**Figure 2 diagnostics-13-02980-f002:**
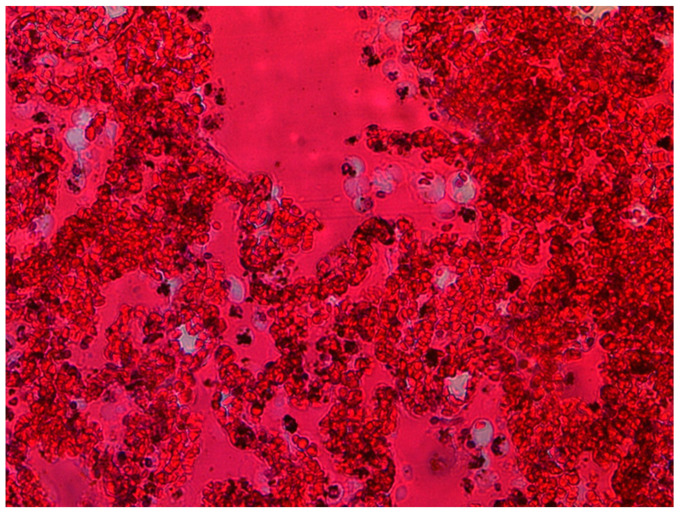
Massive hyperemia, macrophage infiltration (source: Prof. Mirosław Parafiniuk).

**Figure 3 diagnostics-13-02980-f003:**
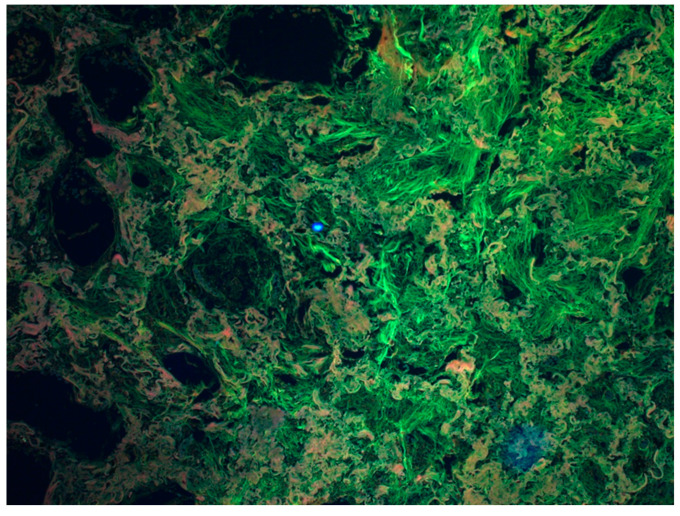
Fibrosis of the lung tissue as a result of a chronic inflammatory process. Visualization using a Zeiss multi-channel microscope (Filter set 25 Zeiss, Excitation TBP 400 + 495 + 570, Emission TBP 460 + 530 + 625). The greenish-gray structures visible in the figure are fibrotic lung tissue. (Source: Prof. Mirosław Parafiniuk).

**Table 1 diagnostics-13-02980-t001:** A summary of genetic tests of lung tissue and nasopharyngeal swabs obtained with postmortem.

Sample	Sex	Age	Method	Lung			Nasopharynx		
				Gene ORF	Gene N	Gene S	Gene ORF	Gene N	Gene S
1	F	2 month		(-)	(-)	(-)	30	30	30
2	F	3 month	RT-PCR	31	31	(-)	28	30	(-)
3	F	14 month		(-)	(-)	(-)	9	9	9
4	F	7		14	14	12	20	18	(-)
5	M	26		9	9	9	(-)	(-)	(-)
6	M	39		37	37	37	(-)	(-)	(-)
7	M	43		25	25	25	(-)	(-)	(-)
8	F	44		20	20	20	(-)	(-)	(-)
9	M	47		34	34	(-)	(-)	(-)	(-)
10	F	51		36	36	(-)	(-)	(-)	(-)
11	M	59		16	16	17	16	17	16
12	M	64		(-)	(-)	(-)	23	22	(-)
13	M	65		26	27	(-)	13	13	(-)
14	M	66		24	24	(-)	24	21	(-)
15	M	67		31	31	33	37	37	37
16	M	70		20	21	27	(-)	(-)	(-)
17	M	70		37	37	(-)	(-)	(-)	(-)
18	M	70		25	25	25	15	15	15
19	M	71		26	26	26	(-)	(-)	(-)
20	F	77		37	37	37	(-)	(-)	(-)
21	F	80		26	26	(-)	(-)	(-)	(-)
22	F	84		15	16	15	15	16	15
23	M	84		28	26	30	(-)	(-)	(-)

F—female (-)—no amplification. M—male.

## Data Availability

Not applicable.

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
