# Peer review of "Molecular Diagnosis of COVID-19 Sudden and Unexplained Deaths: The Insidious Face of the Pandemic"

_diagnostics, 2023, doi:10.3390/diagnostics13182980_

Round 1
Reviewer 1 Report
Dear Authors,
I have read with interest the manuscript "Molecular diagnosis of COVID - 19 sudden and unexplained deaths. The insidious face of the pandemic."
The article is generally interesting and reflects a practical issue represented to the importance of autopsy practice in improving the knowledge of the impact of Covid-19.
However, I detected few issues that must be solved before it can be considered suitable for publication.
Point 1: Abstract.
The Abstract should be completely rewritten according to the Journal's style. You should provide a brief introduction, a description of the matherials and methods, the results, the conclusions.
Point 2: In Line 271, I would suggest to enrich the with the following citations, that highlight the post- mortem persistence of the virus in nasopharyngeal swabs:
Aiello F, Ciotti M, Gallo Afflitto G, Rapanotti MC, Caggiano B, Treglia M, Grelli S, Bernardini S, Mauriello S, Nucci C, Marsella LT, Mancino R. Post-Mortem RT-PCR Assay for SARS-CoV-2 RNA in COVID-19 Patients' Corneal Epithelium, Conjunctival and Nasopharyngeal Swabs. J Clin Med. 2021 Sep 20;10(18):4256. doi: 10.3390/jcm10184256. PMID: 34575369; PMCID: PMC8464749.
Servadei F, Mauriello S, Scimeca M, Caggiano B, Ciotti M, Anemona L, Montanaro M, Giacobbi E, Treglia M, Bernardini S, Marsella LT, Urbano N, Schillaci O, Mauriello A. Persistence of SARS-CoV-2 Viral RNA in Nasopharyngeal Swabs after Death: An Observational Study. Microorganisms. 2021 Apr 10;9(4):800. doi: 10.3390/microorganisms9040800. PMID: 33920259; PMCID: PMC8103507.
Point 3: Please provide detailed statements about funding, conflict of interest, ethical approval and author contribution before the reference list.
Point 4: Correct the style of quotations by putting them in square brackets and not in round brackets, according to the style of the journal.
Point 5: Use the same font size throughout the article.
The English quality is fine, only minor editing is needed.
Author Response
Thank you for all the remarks that improved my manuscript. All changes were provided according to your guidelines.
Reviewer 2 Report
The text discusses the severity of diseases caused by Coronaviridae viruses, particularly the COVID-19 disease caused by SARS-CoV-2. It mentions that while most infections are mild, some can lead to severe respiratory syndromes with high mortality rates. It highlights the initial outbreak in Wuhan, China in December 2019, the subsequent global spread, and the WHO declaring it a pandemic in March 2020. The text also mentions that post-mortem examinations during the pandemic have provided limited information on the disease's pathomechanism and causes of death, especially among children.
The work lacks of "conclusion" section in which you summarize what new did you discover. The study observed variations in virus expression in nasopharyngeal swabs and lung tissue. Some patients had the virus in both, while others had it in only one of them, especially during the acute phase of lung damage. This phenomenon was particularly prominent in children, who have a high likelihood of infection but often exhibit milder symptoms. The low levels of the virus in children could lead to false-negative test results, potentially making them a source of undiagnosed infections.
I think that your discussion is too long and confusing and do not reflects your results. Please re-write. Do you think that SARS-CoV2 is the cause of death or a contributing cause?
Good job, I suggest you improve your discussion by citing some of these articles:
- Autopsy findings in COVID-19-related deaths: a literature review
- Risk Management and Treatment of Coagulation Disorders Related to COVID-19 Infection
- Management of the corpse with suspect, probable or confirmed COVID-19 respiratory infection – Italian interim recommendations for personnel potentially exposed to material from corpses, including body fluids, in morgue structures and during autopsy practice
- COVID-19 Risk Management and Screening in the Penitentiary Facilities of the Salerno Province in Southern Italy
- Myocardial pathology in covid-19-associated cardiac injury: A systematic reviewSARS-CoV-2 and the brain: A review of the current knowledge on neuropathology in COVID-19
Author Response

(The authors gave the same response as above.)

Round 2
Reviewer 1 Report
The manuscript is now suitable for publication.
Thank you